# Genome Investigation and Functional Annotation of *Lactiplantibacillus plantarum* YW11 Revealing Streptin and Ruminococcin-A as Potent Nutritive Bacteriocins against Gut Symbiotic Pathogens

**DOI:** 10.3390/molecules28020491

**Published:** 2023-01-04

**Authors:** Tariq Aziz, Muhammad Naveed, Syeda Izma Makhdoom, Urooj Ali, Muhammad Saad Mughal, Abid Sarwar, Ayaz Ali Khan, Yang Zhennai, Manal Y. Sameeh, Anas S. Dablool, Amnah A. Alharbi, Muhammad Shahzad, Abdulhakeem S. Alamri, Majid Alhomrani

**Affiliations:** 1Beijing Advanced Innovation Center for Food Nutrition and Human Health, Beijing Engineering and Technology Research Center of Food Additives, Beijing Technology and Business University, Beijing 100048, China; 2Institute of Basic Medical Sciences, Khyber Medical University, Peshawar 25100, Pakistan; 3Department of Biotechnology, Faculty of Science and Technology, University of Central Punjab, Lahore 54590, Pakistan; 4Department of Biotechnology, University of Malakand, Chakdara 18800, Pakistan; 5Chemistry Department, Faculty of Applied Sciences, Al-Leith University College, Umm Al-Qura University, Makkah 24831, Saudi Arabia; 6Department of Public Health, Health Sciences College Al-Leith, Umm Al-Qura University, Makkah al-Mukarramah 24382, Saudi Arabia; 7Department of Biochemistry, Faculty of Science, University of Tabuk, Tabuk 71491, Saudi Arabia; 8Department of Clinical Laboratory Sciences, The Faculty of Applied Medical Sciences, Taif University, Taif 21944, Saudi Arabia; 9Centre of Biomedical Sciences Research (CBSR), Deanship of Scientific Research, Taif University, Taif 21944, Saudi Arabia

**Keywords:** *LP* YW11, whole genome, streptin, ruminococcin, nutritive bacteriocin, gut microbiota

## Abstract

All nutrient-rich feed and food environments, as well as animal and human mucosae, include lactic acid bacteria known as *Lactobacillus plantarum*. This study reveals an advanced analysis to study the interaction of probiotics with the gastrointestinal environment, irritable bowel disease, and immune responses along with the analysis of the secondary metabolites’ characteristics of Lp *YW11*. Whole genome sequencing of Lp *YW11* revealed 2297 genes and 1078 functional categories of which 223 relate to carbohydrate metabolism, 21 against stress response, and the remaining 834 are involved in different cellular and metabolic pathways. Moreover, it was found that Lp *YW11* consists of carbohydrate-active enzymes, which mainly contribute to 37 glycoside hydrolase and 28 glycosyltransferase enzyme coding genes. The probiotics obtained from the BACTIBASE database (streptin and Ruminococcin-A bacteriocins) were docked with virulent proteins (cdt, spvB, stxB, and ymt) of *Salmonella*, *Shigella*, *Campylobacter*, and *Yersinia*, respectively. These bacteria are the main pathogenic gut microbes that play a key role in causing various gastrointestinal diseases. The molecular docking, dynamics, and immune simulation analysis in this study predicted streptin and Ruminococcin-A as potent nutritive bacteriocins against gut symbiotic pathogens.

## 1. Introduction

From the genus *Lactobacillus*, *Lactiplantibacillus* is among the top-researched bacterial species. It has been shown to have probiotic and health-promoting benefits and has been accorded the GRAS (generally recognized as safe) status [1]. Due to their capacity to create lactic acid, these bacterial species have been used in food for a long time [2,3]. The most prevalent lactic acid-producing bacterial species in nature, *Lactiplantibacillus*, can be obtained from a broad range of sources, including plant matter, fermented foods (such as yogurt, pickles, cheese), meat products, fruit juice, the gastrointestinal tract of both humans and animals, and wine [4]. Currently, a variety of LAB strains are widely utilized in the food fermentation process and have many uses in medicine, pharmaceuticals, and healthcare. A growing body of research suggests that these bacteria may have health advantages in a variety of intestinal illnesses, including inflammatory bowel disease, due to their demonstrated anti-inflammatory, antibacterial, immune-modulating, and ability to control gut flora activities [5,6,7,8,9]. The probiotic characteristics of these strains are principally responsible for all these activities.

Due to their potential to have health-promoting benefits, probiotics, and products containing them have received a lot of scientific attention [3]. Probiotics are bioactive preparations with substantial concentrations of beneficial natural microorganisms that improve both health and life quality [10]. Human probiotics accounted for more than 90% of the global market in 2021. By 2026, the probiotics market may be worth $3.5 billion globally [11]. *Streptococcus*, *Bifidobacterium*, and enteric bacteria are used as probiotics in food or feed supplements, and “Live biotherapeutic products (LBP)” or “Microecologics for therapeutic application” use enteric bacteria [12,13].

Indirectly improving gut barrier function and directly competing with pathogens to provide antimicrobial effects are the main mechanisms by which these bacteria promote health [14]. Cell surface constituents of *lactobacilli*, such as adhesion molecules, proteins, and polysaccharides [15], influence the adherence of *lactobacilli* to the intestinal mucosa, and these associations may enhance bacterial colonization and host interactions while inhibiting pathogen attachment [16]. Additionally, by causing inhibitors of proinflammatory cytokine production, the anti-inflammatory actions can aid in reducing the host’s local and systemic inflammatory response [10,17]. However, several environmental challenges, including temperature, pH, bile, osmotic, and oxidative stress, are experienced by *lactobacilli* during the fermentation processes.

In contrast to enterotoxigenic species, invasive bacteria damage the host’s epithelial architecture severely. Histological signs of this damage include mucosal ulceration and inflammatory response in the lamina propria. *Salmonella, Shigella, Campylobacter*, invasive *E. coli*, and *Yersinia* are the main pathogens in this group. Although enteric viruses also infiltrate intestinal epithelial cells, they produce much less mucosal damage than invasive bacterial infections do. The natural protective barrier provided by bacteriocins shields LAB from harmful environmental factors and pressures [18]. Additionally, these molecules function as a protective barrier against biofilm development and are essential in cell recognition, attachment to the gut wall, interactions with the immune cells, disruption of pathogen adherence, and contact with cells [19,20]. Bacteriocins have also been linked to probiotics’ immunomodulatory, anticancer, antioxidant, antibacterial, and cholesterol-lowering activities because they have different physiological properties and serve as prebiotics [20,21].

In the food and fermentation industries, bacteriocins can be used as a bio-thickener to increase food viscosity, flavor, and rheological properties. In the cosmetics and pharmaceutical industries, bacteriocins can also be used as a bio-flocculant and bio-absorbent due to their good stability, emulsifying capabilities, and biocompatibility [22]. Due to their ability to produce antimicrobial substances, particularly bacteriocins with antimicrobial activity, *lactobacilli* can also function as natural bio-preservatives [23]. These substances help probiotics survive longer by preventing the growth of pathogenic or hazardous microbes [24]. One of the most significant members of the *lactobacilli*, formerly known as *Lactobacillus plantarum*, has been used extensively as a probiotic. It possesses important probiotic properties, such as immunomodulatory, antioxidant [25], cholesterol-lowering ability, effective nitrate degrading ability, and antimicrobial activity [25,26].

The current study investigates the functional annotation of the genomic components of the *Lp* YW11. The whole genome annotation was conducted with RAST leading to the genomic comparison with other strains of *L. plantarum* via analyzing the ancestral relationship of YW-11. The YW-11 strain was further analyzed for the synthesis and effect of potent bacteriocins on the virulent factors of various gut bacteria via docking analysis. The immune simulations were also determined through C-ImmSimm to analyze the immunomodulatory response of the bacteriocins.

## 2. Results

### 2.1. Genomic Characteristics and Functional Annotation of Lp YW 11

*Lp* YW11 genome consists of 3,181,760 base pairs, with 44.7% GC content, 231 subsystems, and 3065 genes. The genes present in the genome of *Lp* YW11 further comprised 26% sub-system and 74% non-subsystem coverage genes. The subsystem coverage contained 786 genes a total of which 732 genes were characterized and 36 were hypothetical. Similarly, a non-subsystem coverage was comprised of 2297 in a total of which 1196 genes were predicted, and 1101 genes were hypothetical. The total categories of the subsystem were 1078 of which 223 were carbohydrates; 105 were Cofactors, Vitamins, Prosthetic Groups, and Pigments; 46 were cell wall and capsule; 39 were virulence, disease, and defense; 6 were potassium metabolism; 15 were miscellaneous, 9 phages, and plasmid components; 35 membrane transport systems; 5 iron acquisition and metabolism systems; 39 RNA metabolism; 87 nucleosides and nucleotide systems; 134 protein metabolism; 4 cell division and cell cycle; 15 regulation and cell signaling; 4 secondary metabolisms; 48 DNA metabolism; 32 fatty acid biosynthesis; 6 dormancy and sporulation; 16 respiration; 21 stress response; 168 amino acid derivatives; 3 Sulphur metabolism; and 12 phosphorus metabolism (Figure 1). The features of the subsystem along with their functions are described in Appendix A.

The KEGG pathways of the whole genome for the synthesis of essential biomolecules and secondary metabolites, especially exopolysaccharides (EPS), are shown in Appendix A.

### 2.2. Phage Site Prediction

Two significant phage hits were discovered by PHASTER that will increase the genetic variety and provide access to genomic variation as bacteria evolve. The two significant hits were against previously identified phage sections of viruses and other bacterial species (Appendix A).

Figure 2 displays the graphical depiction of the phage areas found by PHASTER in the whole genome of Lp YW11 versus two significant hits. The Lp YW11 genome did not include any clustered regularly interspaced short repeats (CRISPRs), according to CRISPRFinder.

### 2.3. Interpretation of Transporter Proteins

39 proteins were identified as possible transporters from an analysis of the *Lp* YW11 genome, which may aid in several signaling pathways (Appendix A). By tracking signal peptides, the signaling of transporter proteins was investigated. TAT, LIPO, and CS signals given by amino acids at a particular cleavage site were used to predict the predominance of signal peptides (Appendix A). All the anticipated transporter proteins in the *Lp* YW11 genome are found extracellularly, or in the cytoplasm, according to an analysis of their cellular and subcellular localization (Table 1).

### 2.4. Carbohydrate Active Enzyme

The genetic, primary, and biochemical information on carbohydrate-active enzymes (CAZy) that debase, modify, or create glycosidic linkages is examined using the carbohydrate-active enzymes (CAZy) data collection. In the genome of Lp YW11, the CAZy database projected the presence of five primary groups of carbohydrates: glycoside hydrolases, glycosyl transferases, carbohydrate esterase, auxiliary enzymes, and carbohydrate-binding modules (Figure 3). The genome of Lp YW11 contains 82 CAZy genes, of which the glycoside hydrolase family and glycosyltransferase family of enzymes contribute approximately 37 and 28 genes, respectively. This indicates that Lp YW11 is critical for having strong probiotic activity and controlling the immune system against various pathogens (Appendix A). The identified genes for the coding of carbohydrate coding enzymes were further annotated by dbCAN which analyzed HMMER, DIAMOND, and CGC regions in the whole genome of Lp YW11 (Appendix A).

### 2.5. EPS-Producing Genes in Lp YW11

In the genome of *Lp* YW11, antiSMASH predicted four key regions that create EPS, namely region 1 (cyclic lactone autoinducer), region 2 (terpene), region 3 (T3PKS), and region 4 (Ripp similar) (Figure 4). Comparative EPS-producing areas of and other *Lactiplantibacillus plantarum* genomes are shown in Figure 5.

### 2.6. Interaction of EPS-Producing Genes

The 43 identified genes involved in the production of EPS were analyzed for their functioning regarding immune response generation and enhanced improvement in intestinal bowel diseases. The string network of 43 identified genes in the production of EPS is shown in Figure 6. The pathways involved in immune stress response and improvement in intestinal bowel diseases are D-alanine metabolism, folate biosynthesis, fatty acid biosynthesis, one carbon pool by folate, glycerophospholipid metabolism, peptidoglycan biosynthesis, biotin metabolism, pyrimidine metabolism, amino sugar and nucleotide sugar metabolism, purine metabolism and aminoacyl-tRNA biosynthesis (Appendix A). These pathways are involved in immune stress response and IBD improvement at gene and nucleotide levels. Any change in the micro-genetic level causes abrupt changes in the production of the essential enzyme that ultimately causes alterations in the whole metabolic pathways.

### 2.7. Comparative Genome Analysis

Genome collinearity analysis may help in identifying and understanding genetic differences between analyzed and reference genomes due to inversion, transposition, and other phenomena, as well as exhibiting the insertion, excision, and other details of sequences. It was observed that *Lp* YW11 had a high collinearity relationship with HC-2, LLY-606, pc-26, TMW, and WLPL04 strains of *Lactiplantibacillus plantarum* (Figure 7) (Appendix A).

Further, the multiply aligned genomes of *Lactiplantibacillus plantarum* showed the alignment of EPS-producing, transporter, and regulatory genes have been shown in Figure 8A, which corresponds to the enhanced capabilities of *Lp* YW11 as compared to other strains. The phylogenetic tree of *Lp* YW11 was constructed based on sequence similarity with HC-2, LLY-606, pc-26, TMW, and WLPL04 strains of *Lactiplantibacillus plantarum* genomes which showed the least close relationship with the genomes of comparison has been shown in Figure 8B.

### 2.8. Molecular Modelling

With a QMEAN Z-Score of 0.724, 0.785, 0.760, and 0.876 independently, the trRosetta predicted the tertiary design of cdt, spvB, stxB, and ymt with 81%, 96%, 81%, and 9% grouping personalities. The protein structure had a higher certainty expectation, as shown by the scoring list and other pertinent criteria. The trRosetta-determined quality rating was within both the ideal and standard range. As a result, the building met quality standards and was important for the use of docking studies and other objectives. Figure 9 shows the projected tertiary construction of cdt, spvB, stxB, and ymt.

The trRosetta anticipated the tertiary design of streptin and Ruminococcin-A with a QMEAN Z-Score of 0.196 and 0.214, respectively (Figure 10).

#### 2.8.1. Interactome Prediction

The results of building the interactome of the proteins cdt, spvB, stxB, and ymt revealed that the proteins of interest associate with various other proteins, many of which are restricted to the nucleus, plasma membrane, and others of which are located in extracellular space. Following the creation of the interactome using STRING on the web device, only the relationships with the highest levels of certainty were chosen for further analysis in this review, and associations with medium and low levels of certainty were ignored (Figure 11).

#### 2.8.2. Molecular Docking and Dynamic Simulations

The streptin and ruminococcin-affinity A’s for the gut bacterial pathogenic genes cdt, spvB, stxB, and ymt were examined using HDock. Streptin’s strongest interaction energies were −216.97 with cdt (Figure 12A), −221.42 with spvB (Figure 12B), −179.47 with stxB (Figure 12C), and −122.43 with ymt (Figure 12D). Similar to this, the best binding scores for Ruminococcin-A with cdt, spvB, stxB, and ymt were −223.05 for cdt, −233.62 for spvB, −182.05 for stxB, and −270.65 for ymt (Figure 12E–H).

The stability of the docked complexes was predicted by the molecular simulation run using iMODs, as illustrated in Figure 13 and Figure 14, based on field force. The bacteriocins and pathogenic genes interacted most steadily on 500 ns, according to the analysis of the docked complex’s deformability potential. The estimated B-factor supported this prediction, and the eigenvalues showed how much energy was needed to deform the complex with little variation. Little to no confusing interactions were expected from the docked complex’s covariance and elastic maps.

#### 2.8.3. Immune Simulations

The developed vaccine received a fantastic immunological response from C-ImmSim. Appendix A show the immune cell population for each consecutive injection day, including B-cell, PLB, and Th-cell populations These figures show that throughout the bacteriocins, active and resting memory B cells along with the IgM isotypes are continuously produced and replicate exponentially. IgM + IgG isotypes, as well as isolated IgM and IgG1 isotypes, increase exponentially in the PLB graph during the first week of probiotic use. The Th-cell graph demonstrates the dynamic presentation of MHC-II molecules in the first five days after absorption.

## 3. Discussion

Since many *Lactiplantibacillus plantarum* have a long history of use in food due to the lactic corrosive delivering bacterium, there is growing interest in it. *Lactiplantibacillus plantarum* is one of the most extensively studied bacterial species belonging to the class *Lactobacillus* with demonstrated probiotic and well-being advancing impacts and has been “by and large perceived as protected” (GRAS) status. Numerous studies have shown that they are capable of enhancing important medical benefits, such as suppressing the growth of microorganisms by demonstrating immunomodulatory effects, directing stomach greenery, altering the gastrointestinal microbiota, improving resistance, advancing processing, safeguarding gastric mucosa, and treating gastrointestinal contaminations, incendiary entrails infections, and unfavorably susceptible diseases [27]

According to a genome-wide investigation of *Lp* YW11, the genome is 3,181,760 bp in size, with 26% of its genes covered by subsystems and 74% by non-subsystem genes. The genes involved in sub-system coverage are involved in the creation of beneficial metabolites as well as numerous metabolic pathways. By taking into account two significant phage areas against viruses and other bacterial species, the genetic diversity in *Lp* YW11 was examined. Potential transporter proteins that were engaged in multiple signaling pathways were searched for in the *Lp* YW11 genome. The TAT, LIPO, and CS signals, which were provided by particular amino acids at a particular cleavage location, were used to predict the signal peptide. The projected transporter proteins’ cellular and subcellular locations were also examined, and the results demonstrated that all transporter proteins were found extracellularly. The glycoside hydrolase and glycosyltransferase families of carbohydrate-active enzymes, which are largely contributed to by the genome of *Lp* YW11, are crucial for significant probiotic activity and the control of the immune response to a variety of pathogens.

The current study also indicates the prediction of exopolysaccharide areas, including terpene, T3PKS, and RiPP-like regions, in the genome of *Lp* YW11. The abundance of EPS production in our query sequence was shown by comparing the *Lp* YW11 with other bacterial species such as enterococcus, bacillus cereus, and halobacillus. The 43 EPS-producing genes were further examined for their role in immune response pathways, protein-protein interactions, and their improvement of inflammatory bowel disease (IBD) via STRING network creation. The STRING analysis revealed several pathways, including those involved in purine, biotin, pyrimidine, folate, and peptidoglycan metabolism as well as immune response and IBD improvement.

With the *Lactiplantibacillus plantarum* strains HC-2, LLY-606, pc-26, TMW, and WLPL04, a comparative genomic analysis of *Lp* YW11 was conducted. Concerning comparator genomes, the *Lp* YW11 displayed the highest collinearity and least ancestral relationship. These in-depth interpretations offer opportunities to modify or redesign this strain to combine its properties in the human body. In the gastrointestinal tract of humans, some pathogenic bacteria are causing various pathogenic diseases. These bacteria mainly include *Salmonella*, *Shigella*, *Campylobacter*, and *Yersinia* which are highly resistant to modern antibiotics. Probiotics-producing bacteria are of great interest to overcome this problem. In the current study, we have discussed two bacteriocins secreted by YW-11 retrieved from BACTIBASE i.e., streptin and Ruminococcin-A. These bacteriocins were analyzed against the virulent genes i.e., cdt, spvB, stxB, and ymt of *Salmonella, Shigella, Campylobacter,* and *Yersinia*, respectively.

The docking scores, molecular simulations, and immune simulations have shown that the probiotics produced by *Lp* YW-11 are safe to use against drug-resistant pathogenic gut bacteria and have enhanced nutritive properties. In addition, the usage of *Lp* YW11 in modern food products can be improved by using its current metabolic and biochemical pathways to identify and direct similar strains as potential probiotics in foods and beverages.

## 4. Materials and Methods

### 4.1. Bacterial Strain and Culture Condition

Previously isolated *Lactiplantibacillus plantarum* YW11 (*Lp* YW11), available on GenBank with ID number: CP035031.1, was kept in frozen (−9 °C) stocks of MRS broth that had been provided with 20% (*v*/*v*) glycerol. The strains were identified using different tests such as the gram reaction, cell morphology, and catalase assays. The 16S rDNA sequencing analysis and the API 50 CHL test (bio-Merieux, Craponne, France) were used to identify the strains [28,29,30,31,32,33,34].

### 4.2. DNA Extraction and Whole Genome Sequencing

The Wizard^®^ Genomic DNA Purification Kit from Promega was used to extract genomic DNA, and the TBS-380 fluorometer from Turner Bio Systems Inc., (Sunnyvale, CA, USA), was used to measure the amount of DNA. For additional examination, high-quality DNA (OD260/280 = 1.8–2.0, >20 ug) was employed. Using the NEXTflexTM Rapid DNA-Seq Kit, Illumina sequencing libraries were created from the sheared fragments. Using the NEXTflexTM Rapid DNA-Seq Kit, Illumina sequencing libraries were created from the sheared fragments. Briefly, end-repair and phosphorylation of the 5′ prime ends came first. The 3′ ends were then capped with an A-tail and ligated to sequencing adapters. The last stage was utilizing PCR to enrich the adapters-ligated products. On an Illumina HiSeq X Ten platform, the generated libraries were used for paired-end Illumina sequencing (2150 bp). Single Molecule Real-Time (SMRT) technology and Illumina sequencing platforms were used to complete the full genome sequence of the chosen strain (*Lp* YW11), which has the accession number (GCA 004028295.1) [35]. The complexity of the genome was evaluated using the Illumina data. The Qiagen DNA extraction kit was used to isolate the genomic DNA following the manufacturer’s instructions.

### 4.3. Genomic Investigation

Rapid annotation of using subsystem technology (RAST) (https://rast.nmpdr.org/) accessed on 10 October 2022, foresaw the utilitarian comment of qualities connected with various cell and metabolic pathways. The PHASTER web server (https://phaster.ca/) accessed on 10 October 2022, predicted the prophage regions in the *L. plantarum* YW-11 genome. CRISPRFinder (available at https://bioinformaticshome.com/devices/DNA-successioninvestigation/portrayals/CRISPRFinder.html) accessed on 12 October 2022 identified clustered regularly interspaced short palindromic repeats.

Further, the arrangement comparability search was performed by NCBI BLAST (https://blast.ncbi.nlm.nih.gov/Blast.cgi) accessed on 15 October 2022. The Transporter Classification Database (TCDB) was utilized to examine possible carriers from Lp 13-3 (https://tcdb.org/) accessed on 16 October 2022 which were oppressed for their flagging abilities by Signal Peptide (SignalP 5.0) (https://services.healthtech.dtu.dk/service.php?SignalP-5.0) accessed on 17 October 2022.

### 4.4. Genome Comparison

The Seed Viewer of RAST (https://rast.nmpdr.org/) accessed on 19 October 2022 was used to compare the collinearity of the *Lactiplantibacillus plantarum* strains YW11, HC-2, LLY-606, pc-26, TMW, and WLPL04. MAUVE Alignment (http://darlinglab.org/mauve/mauve.html) accessed on 20 October 2022, which primarily evaluates the evolutionary history of comparative genomes, conducted the multiple genome alignment of the genes. Additionally, CV-Tree (http://cvtree.online/v3/cvtree/) accessed on 22 October 2022 created a phylogenetic analysis utilizing the neighbor-joining method for comparative genome analysis.

### 4.5. Functional Annotation

Glycoside hydrolases (GHs) are sugar dynamic chemicals (CAZy), which were anticipated by the CAZy data collection (http://www.cazy.org/) accessed on 23 October 2022 and further defined by the DbCAN meta server (https://bcb.unl.edu/dbCAN2/) accessed on 24 October 2022. To anticipate any antibiotic-resistant genes in the *L. plantarum* YW-11 genome, the identification of antibiotic resistance factors CARD (Comprehensive Antibiotic Resistance Database) (http://arpcard.mcmaster.ca/) accessed on 25 October 2022 was also conducted. STRING predicted the cryptoscopic protein-protein connections of annotated gene-creating properties.

### 4.6. Bacteriocin Production

PathogenFinder (http://cge.cbs.dtu.dk/services/PathogenFinder/) accessed on 26 October 2022 was used to examine the bacterial pathogenicity and determine if *L. plantarum* YW-11 was a pathogen or not. The BACTIBASE database was used to screen the nucleotide sequence of *L. plantrum* YW-11. The antiSMASH bacterial variation was used to analyze the exopolysaccharide biosynthesis quality clusters throughout *L. plantarum* YW-11’s whole genome (https://antismash.secondarymetabolites.org/#!/begin) accessed on 27 October 2022.

### 4.7. Molecular Modelling

The virulent factors of four main pathogenic gut bacteria were analyzed from VFDB. The targeted pathogenic gut bacteria were *Campylobacter jejuni, Salmonella enterica, Shigella dysenteriae,* and *Yersinia pestis* along with their virulent genes, i.,e., cdt, spvB, stxB, and ymt, respectively. Regarding bacteriocins, two main bacteriocins of *Lactiplantibacillus plantarum* were retrieved from BACTIBASE, namely, streptin and Ruminococcin-A. All retrieved sequences, i.e., of virulent genes and bacteriocins were modeled using trRosetta.

### 4.8. Interactome Prediction of Pathogenic genes of Gut Bacteria

The interactome for the proteins cdt, spvB, stxB, and ymt was predicted using the Search Tool for the Retrieval of Interacting Genes (STRING), which provides a database of known and anticipated protein interactions for 2 million proteins. We were able to identify the structure, interactome, and functions of proteins from *Campylobacter jejuni*, *Salmonella enterica*, *Shigella dysenteriae*, and *Yersinia pestis* that interact with other proteins using our web tool.

### 4.9. Molecular Docking and Dynamic Simulations

HDock is capable of anticipating the preferred orientation in which two molecules will bind to form a stable compound. After that, scoring formulas can be used to calculate the binding affinity or the strength of a molecule’s connection to another. To see the docking, Pymol 3D Molecular Structure was employed. For the simulations of the docked complexes, the online web program iMODS was used.

### 4.10. Immune Simulations

On the C-ImmSim server (http://kraken.iac.rm.cnr.it/C-IMMSIM/) accessed on 30 October 2022, the sequence was performed to confirm the immunological response elicited by the vaccine design. Based on specifics such as injection timing, the server delivers immunological activity against bacteriocins or any medicine. It displays the bacteriocin response of B and T lymphocytes and predicts the bacteriocin response of immunoglobulins and immunocomplexes.

## 5. Conclusions

The current study predicted the exopolysaccharides of *Lp YW11*, including terpene, T3PKS, and RiPP-like regions. The abundance of EPS production in the genome was further validated by comparing the information with other bacterial species such as *enterococcus*, *bacillus cereus*, and *halobacillus*. Among the identified bacteriocins, two (streptin and Ruminococcin-A) were further analyzed for their probiotic role through docking with virulent proteins of pathogenic bacterial species. The docking, molecular dynamics, and immune simulation analysis showed that both the bacteriocins are potent inhibitors of the target bacterial pathogens and help the human host elicit a strong immune response against the pathogenic bacteria. These findings elucidate that these bacteriocins can be utilized in food safety approaches, and the *Lp YW11* producing these bacteriocins can be used in food starter cultures.

## Figures and Tables

**Figure 1 molecules-28-00491-f001:**
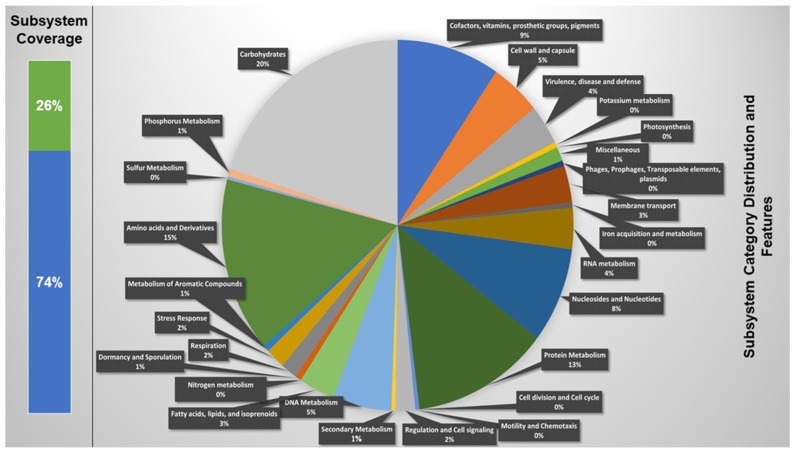
Subsystem coverage and distribution of *Lp* YW11 genome by RAST.

**Figure 2 molecules-28-00491-f002:**
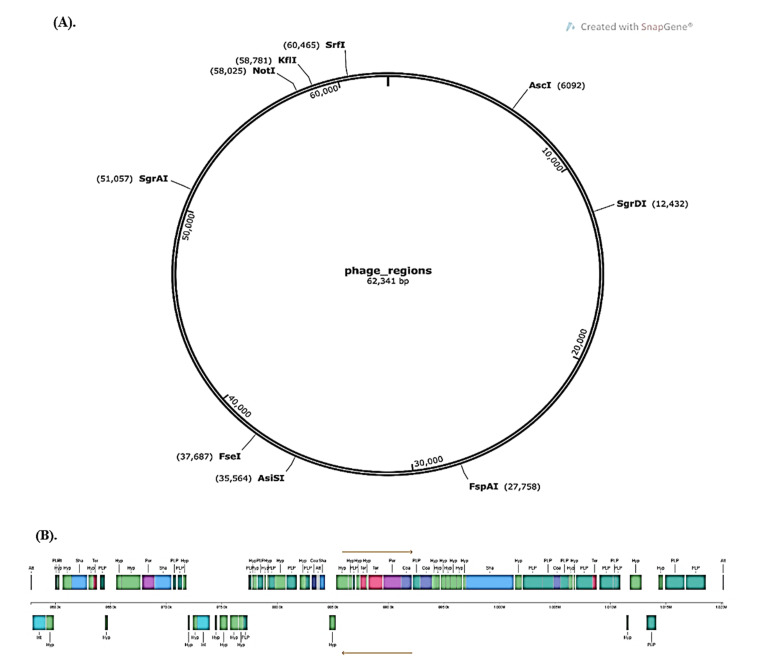
PHASTER’s prediction of phage sites (**A**) Complete genome prediction of *Lp* YW11 with phage areas in the lactoplantibacillus *plantarum* genome (**B**) Expanded view of the genome with phage sites.

**Figure 3 molecules-28-00491-f003:**
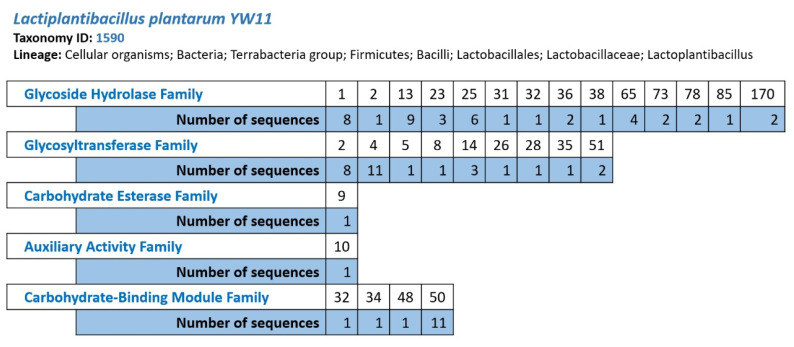
The prediction of the glycoside hydrolase family, glycosyltransferase family, carbohydrate esterase family, auxiliary esterase family, and carbohydrate-binding module family by the CAZy database.

**Figure 4 molecules-28-00491-f004:**
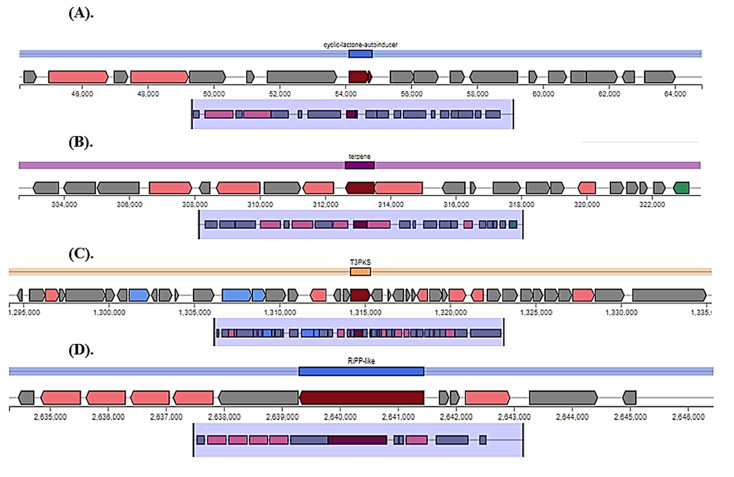
Shows the *Lp* YW11 genome’s EPS-producing sites. Core biosynthetic genes are represented by the colors red, pink, green, blue, and grey. Regulatory genes are represented by the color green (Other genes). (**A**) The autoinducer cyclic lactone (**B**) terpene (**C**) T3PKS (**D**) RiPP likes.

**Figure 5 molecules-28-00491-f005:**
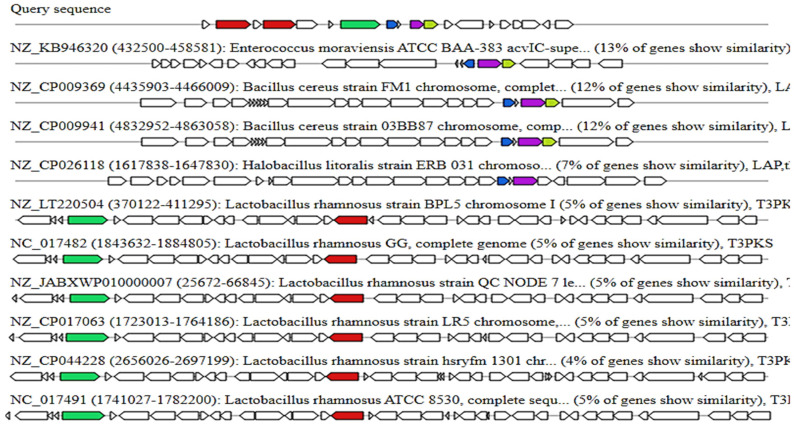
Prediction of Exopolysaccharides (EPS) in red by antiSMASH along with the genomic comparison of *Lp* YW11 with other strains of *Lactiplantibacillus plantarum*.

**Figure 6 molecules-28-00491-f006:**
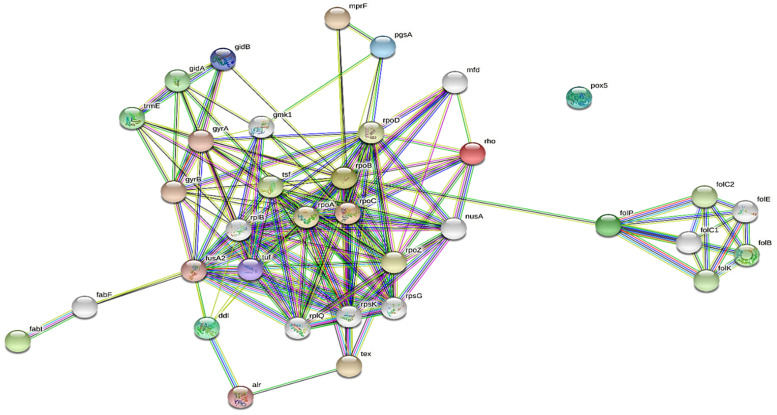
STRING network of 43 genes involved in EPS production that causes immune stress response and intestinal bowel disease improvement.

**Figure 7 molecules-28-00491-f007:**
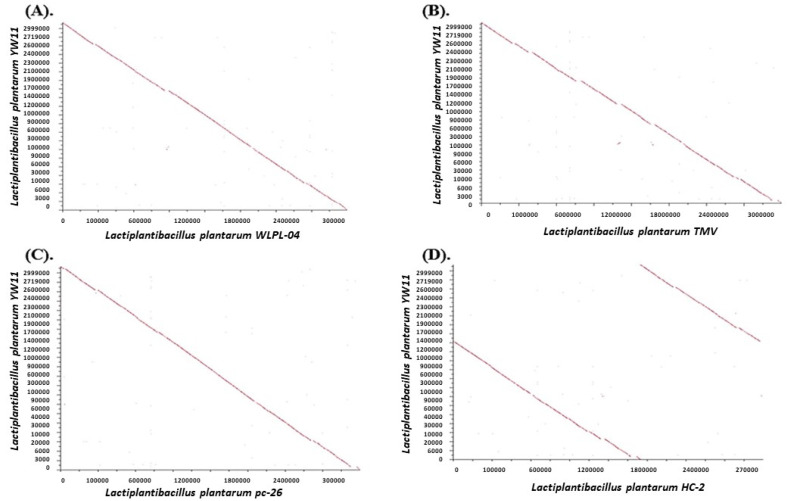
Collinearity relationship of YW-11 with (**A**) WLPL04, (**B**). TMW, (**C**) pc-26 and (**D**) HC-2, strains of *Lactiplantibacillus plantarum* using SEED Viewer of RAST.

**Figure 8 molecules-28-00491-f008:**
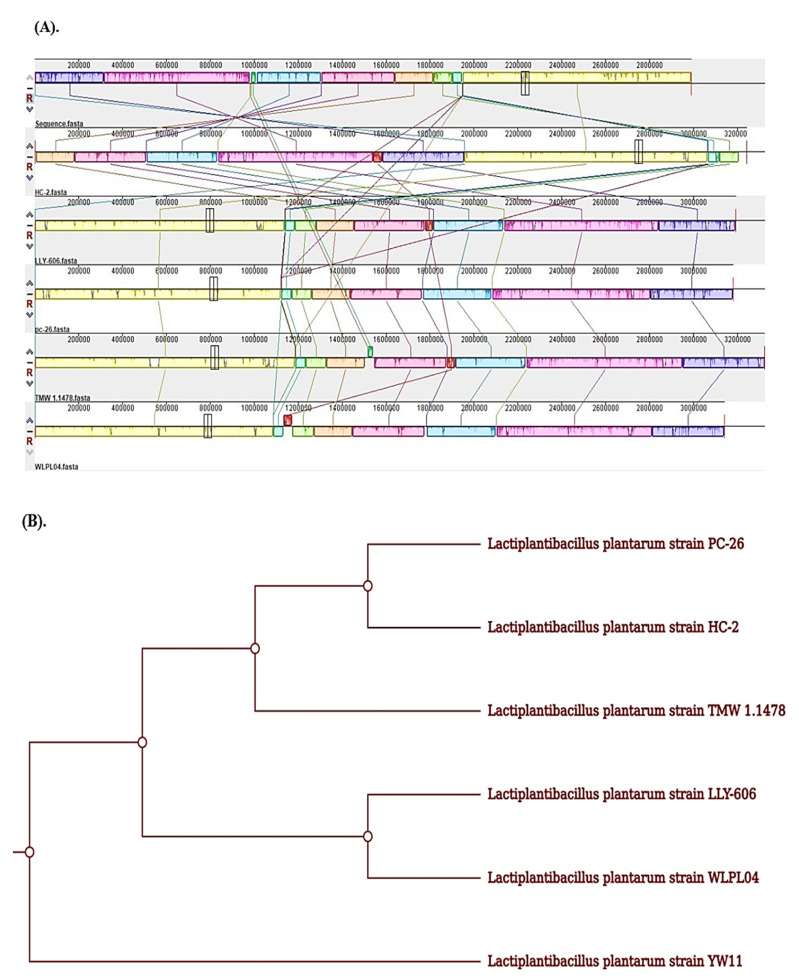
(**A**) Collinearity relationship of *Lp* YW11 with HC-2, LLY-606, pc-26, TMW, and WLPL04 showing aligned EPS-producing genes (Pink), transporter genes (Blue), regulatory genes (Green). (**B**) Phylogenetic Analysis of *Lp* YW11 with HC-2, LLY-606, pc-26, TMW and WLPL04.

**Figure 9 molecules-28-00491-f009:**
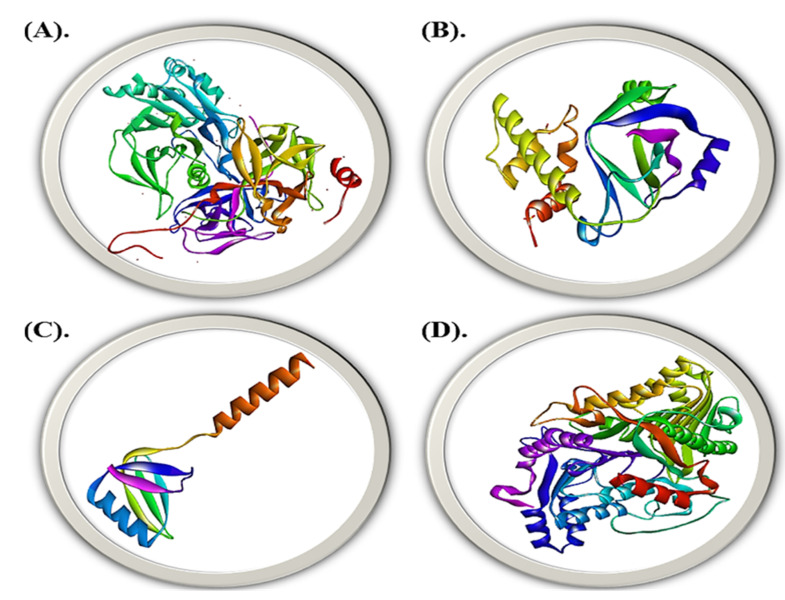
3-D structure prediction of the virulent genes of gut bacteria via trRossetta. (**A**) cdt, (**B**) spvB, (**C**) stxB, and (**D**) ymt.

**Figure 10 molecules-28-00491-f010:**
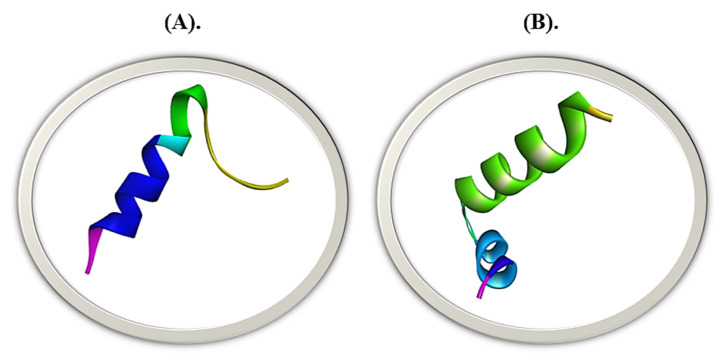
3-D modeled structures of bacteriocins via trRosetta (**A**) Streptin (**B**) Ruminococcin-A.

**Figure 11 molecules-28-00491-f011:**
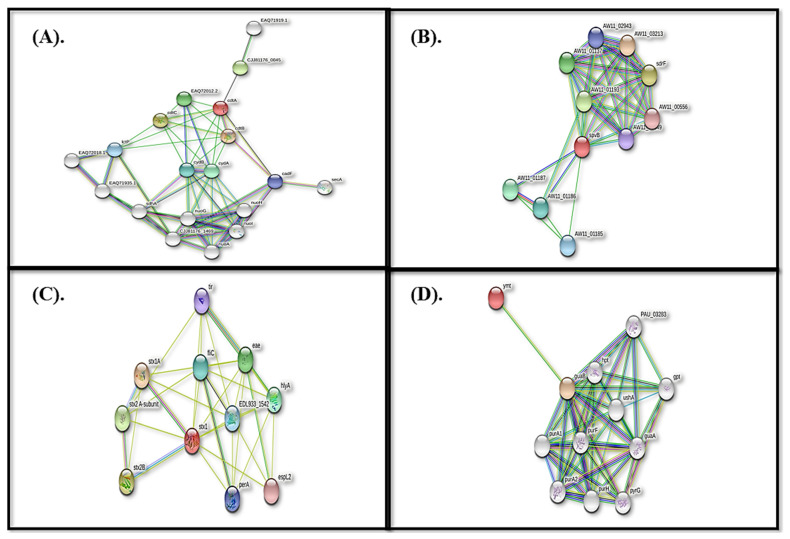
Interactome prediction through STRING. (**A**) ctd (**B**) spvB (**C**) stxB and (**D**) ymt.

**Figure 12 molecules-28-00491-f012:**
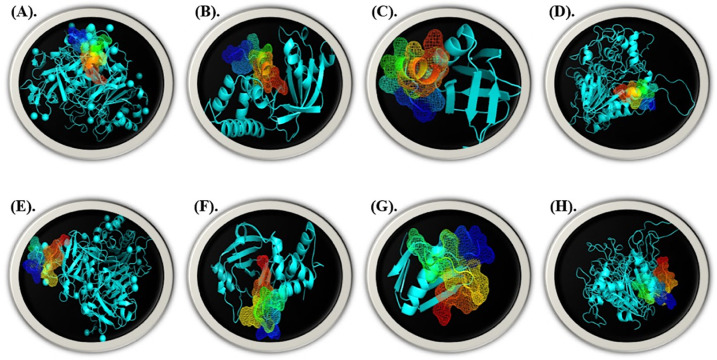
Visualized docked complexes by Pymol of streptin with (**A**) cdt (**B**) spvB (**C**) stxB (**D**) ymt and of Ruminococcin-A with (**E**) cdt (**F**) spvB (**G**) stxB (**H**) ymt.

**Figure 13 molecules-28-00491-f013:**
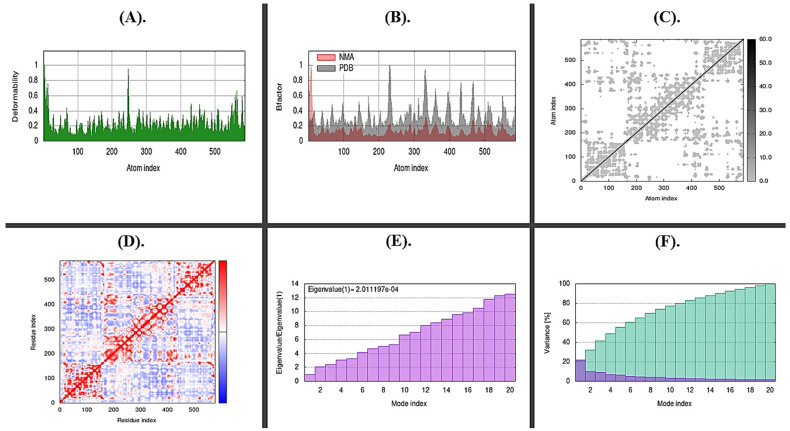
Shows a simulation of molecular dynamics of the best docked complex of streptin with spvB. (**A**) Deformability of the complex; (**B**) B-Factor graph, the plot indicates a comparative PDB plot, but there are no validated structures of our molecules on PDB; (**C**) Elastic Network (Grey matter indicates stiffer region); (**D**) Covariance Map: correlated (red), uncorrelated (white), or anti-correlated (blue) motions; (**E**) The eigenvalue plot illustrates the minimum energy required to deform the complex; (**F**) Variance individual variance (purple) and cumulative variance (green).

**Figure 14 molecules-28-00491-f014:**
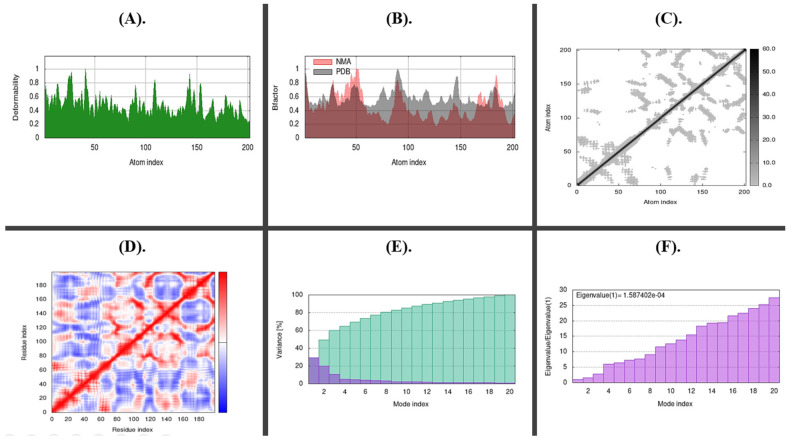
Shows a simulation of molecular dynamics of the best-docked complex of Ruminococ-cin-A with ymt. (**A**). Deformability of the complex; (**B**). B-Factor graph, the plot indicates a comparative PDB plot, but there are no validated structures of our molecules on PDB; (**C**). Elastic Network (Grey matter indicates stiffer region); (**D**). Covariance Map: correlated (red), uncorrelated (white), or anti-correlated (blue) motions; (**E**). Variance individual variance (purple) and cumulative variance (green); (**F**). The eigenvalue plot illustrates the minimum energy required to deform the complex.

**Table 1 molecules-28-00491-t001:** Interpretation of cellular and sub-cellular localization through CELLO of predicted transporter proteins by TCDB.

TCDB ID	Protein Name	CELLO
Q13U92	Putative membrane-anchored cell sub-family protein	Extracellular
WP_060331514.1	autotransporter-associated beta structure protein	Extracellular
CAH35630.1	putative outer membrane protein	Extracellular
Q79FV6	PE-PGRS FAMILY PROTEIN	Extracellular
Q86IX4	Nsp1_C domain-containing protein	Extracellular
WP_010035418.1	FG-GAP repeat protein	Extracellular
B1Z6U1	Filamentous haemagglutinin family protein	Extracellular
B3FNS7	Trimeric autotransporter adhesin protein	Extracellular
XP_021194905.1	papilin isoform X5	Extracellular
Q86AS3	EGF-like domain-containing protein	Extracellular
P46531.4	Neurogenic locus notch homolog protein	Extracellular
A2VEC9.2	SCO-spondin protein	Extracellular
P06620.1	Ice nucleation protein	Extracellular
G0SBQ3.1	Nucleoporin NSP1	Extracellular
B0FXJ3	Ice nucleation protein	Extracellular
WP_057332199.1	PE family protein	Extracellular
L7VCL1	PE-PGRS family protein	Extracellular
P10079	Fibropellin-1	Extracellular
Q03650	Cysteine-rich, acidic integral membrane protein	Extracellular
G0SAK3	Nucleoporin NUP145	Extracellular
WP_084871275.1	carbohydrate-binding domain protein	Extracellular
O00468	Agrin	Extracellular
D3KYQ3	Macronuclear nucleoporin protein	Extracellular
L7V457	PE-PGRS family protein	Extracellular
A7SCE9	Predicted protein (Fragment)	Extracellular
P37198	Nuclear pore glycoprotein	Extracellular
E1ZJE6	Putative uncharacterized protein	Extracellular
WP_010044366.1	hypothetical protein	Extracellular
P35658	Nuclear pore complex protein	Extracellular
Q749L8	Cytochrome c	Extracellular
WP_010044253.1	hypothetical protein	Extracellular
AVM72784.1	magnetosome membrane-specific protein	Extracellular
A8CG34	Nuclear envelope pore membrane protein	Extracellular
A6NF01.2	Putative nuclear envelope pore membrane protein	Extracellular
P71187	TrbL protein	Extracellular
Q8H384	Cadmium selective transporter protein	Extracellular
A3M3H0	Adhesin Ata autotransporter	Extracellular
MBC7395593.1	flagellar hook-length control protein	Extracellular
XP_003062523.1	predicted protein	Extracellular

## Data Availability

Not applicable.

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
