# Peer review of "Genome Investigation and Functional Annotation of Lactiplantibacillus plantarum YW11 Revealing Streptin and Ruminococcin-A as Potent Nutritive Bacteriocins against Gut Symbiotic Pathogens"

_molecules, 2023, doi:10.3390/molecules28020491_

Round 1

Reviewer 1 Report

Review comments

The study is based on the Genome investigation and Functional Annotation of Lacti-2 plantibacillus plantarum YW11 Revealing Streptin and Ruminococcin-A as Potent Nutritive Bacteriocins against Gut Symbiotic Pathogens using advanced analysis. The work seems very interesting and the bacteria has high potential for future implementation. Is there any novelty of this work except the strain? The major focus of the manuscript is the detection of two bacteriocins in the YW11 strain, but the detailed methodology and the obtained results from the BACTIBASE database are unclear.

The authors analyze the genome investigation and functional annotation of Lactiplantibacillus plantarum YW11 for strepin and ruminococcin-A as potent nutritive bacteriocins against gut symbiotic pathogens where different bioinformatics tools have been used. In my opinion most of the figures have been copy and pasted from the program directly which is not convincing. In addition, figure1 and figure 2 don’t give information clearly. Omitting figure 1 is good from the main manuscript and shift to the supplementary data. It is surprising to see how the authors used such a vague diagram obtained from the KEGG pathway in Figure 2 to represent some metabolic pathways, the figure lacks details, interpretation and comprehensiveness for the readers. Similar aspects are related to other figures that have been directly copied and pasted. It is recommended to make their own figures and pathways. The information provided about virulence factor through VFDB is not convincing to me. Even in the supplementary figures, there have been so many figures have been directly snapping out from the KEGG pathway which I found not clear. The results obtained from various genome analysis tools have been directly snipped and inserted in the manuscript but there are no any explanation or interpretation of the data obtained. There is no sufficient discussion about the obtained data.

It would be more reliable if the functional annotation were predicted using standalone tools like Prokka annotation, EggNOG database, and even for the prediction of virulence factors. So, I think the manuscript would be good if it were well written once more to support the results of the In-silico analysis.

The manuscript needs some minor corrections that are highlighted below and needs to be done.

Minor corrections

1.      Needs English corrections and more precise scientific writing.

2.      In page no 2 of the introduction section, line no 53, there should be species instead of specie. 

3.      In page no 2, line no 83 and 84, all the genus name should be in italics in scientific writing. So please correct the mistake e.g.; Salmonella, Shigella, Campylobacter etc. +Similar types of pattern in the whole manuscript was observed during the reading so it needs to be changed like line no 182, line no 183, line no 192, line no 193.

4.      In page no 3, line no 143 and 145, please check the line spacing. Besides that, please check double spacing in line no 371, line no 392, line no 395 and line no 397.

5.      In the discussion section, “The glycoside hydrolase and glycosyltransferase families of carbohydrate-active enzymes, which are largely contributed to by the genome of Lp YW11, are crucial for significant probiotic activity and the control of the immune response to a variety of pathogens.” Please explain how the GH and GT families present in YW11 strain are crucial for probiotic activity.

Author Response

Dear Editor and Reviewers,

We thank you for your critical review of our manuscript and for giving us the chance to submit the revision. The comments were constructive, and we tried to address all of them. We have attached the revised manuscript and a specific response to all three reviewers. We also highlighted the changes in the manuscript. We hope that the manuscript will now be considered suitable for publication.

Reviewer 1

The study is based on the Genome investigation and Functional Annotation of Lactiplantibacillus plantarum YW11 Revealing Streptin and Ruminococcin-A as Potent Nutritive Bacteriocins against Gut Symbiotic Pathogens using advanced analysis. The work seems very interesting, and the bacteria has high potential for future implementation. Is there any novelty of this work except the strain? The major focus of the manuscript is the detection of two bacteriocins in the YW11 strain, but the detailed methodology and the obtained results from the BACTIBASE database are unclear.

Response: Thank you very much for the appreciation. Yes, the functional annotation of Lactobacillus plantarum YW-11 and analysis of potential bacteriocins, i.e., streptin and Ruminococcin-A against pathogenic gut bacteria, describes the novelty of this study. Further, the BACTIBASE database was used to retrieve the data of bacteriocins present in the annotated genome of YW-11. We have used this strain in our different studies which has been cited already in the  manuscript as this strain is capable of producing EPS and CLA Isomer i.e. Rumenic acid and its different bioactivities has been well studied. The novelty of this work is we found these two bacteriocins streptin and Ruminococcin-A against pathogenic gut bacteria through different databases via in-silico /molecular docking studies.

The authors analyze the genome investigation and functional annotation of Lactiplantibacillus Plantarum YW11 for strepin and ruminococcin-A as potent nutritive bacteriocins against gut symbiotic pathogens where different bioinformatics tools have been used. In my opinion most of the figures have been copy and pasted from the program directly which is not convincing. In addition, figure1 and figure 2 don’t give information clearly. Omitting figure 1 is good from the main manuscript and shift to the supplementary data. It is surprising to see how the authors used such a vague diagram obtained from the KEGG pathway in Figure 2 to represent some metabolic pathways, the figure lacks details, interpretation and comprehensiveness for the readers. Similar aspects are related to other figures that have been directly copied and pasted. It is recommended to make their own figures and pathways. The information provided about virulence factor through VFDB is not convincing to me. Even in the supplementary figures, there have been so many figures have been directly snapping out from the KEGG pathway which I found not clear. The results obtained from various genome analysis tools have been directly snipped and inserted in the manuscript but there are no any explanation or interpretation of the data obtained. There is no sufficient discussion about the obtained data.

Response: Thank you so much for your comment. The graphics quality of figures 1, 2, and all supplementary figures have been improved. Figure 1 can be part of supplementary data but, it would be more suitable if it will remain part of the manuscript text as it is the graphical representation of the annotated genome. The tabular data of the annotated functions of the genome have been adjusted in the supplementary data, so it would be more convincing if Figure 1 (the graphical representation of the functional annotation) were part of the main manuscript data.

It would be more reliable if the functional annotation were predicted using standalone tools like Prokka annotation, EggNOG database, and even for the prediction of virulence factors. So, I think the manuscript would be good if it were well written once more to support the results of the In-silico analysis.

Response: Thank you so much for your comment. The suggestion regarding Prokka annotation is considerable. The Prokka and RAST seed viewers have no significant difference regarding the functional annotation of prokaryotic genomes. Both provide information regarding systems and sub-systems of the genome in tabular and graphical representation. Therefore, while the standalone tools are no doubt preferred, the alternate online software provides data of comparable quality and is much easier to use than the standalone versions.

Minor corrections

1. Needs English corrections and more precise scientific writing.

Response: Thank you for your comment. Proofreading of the revised manuscript is done thoroughly through a native speaker. Please see the revised manuscript.

2. In page no 2 of the introduction section, line no 53, there should be species instead of specie.

Response: Thank you for your comment. It has been corrected in the revised manuscript.

3. In page no 2, line no 83 and 84, all the genus name should be in italics in scientific writing. So please correct the mistake e.g.; Salmonella, Shigella, Campylobacter etc. +Similar types of pattern in the whole manuscript was observed during the reading so it needs to be changed like line no 182, line no 183, line no 192, line no 193.

Response: Thank you for your comment. The corrections have been done accordingly in the revised manuscript.

4. In page no 3, line no 143 and 145, please check the line spacing. Besides that, please check double spacing in line no 371, line no 392, line no 395 and line no 397.

Response: Thank you for your comment. The corrections have been done accordingly in the revised manuscript.

5. In the discussion section, “The glycoside hydrolase and glycosyltransferase families of carbohydrate-active enzymes, which are largely contributed to by the genome of Lp YW11, are crucial for significant probiotic activity and the control of the immune response to a variety of pathogens.” Please explain how the GH and GT families present in YW11 strain are crucial for probiotic activity.

Response: Thank you for your comment. The carbohydrate-active enzymes of Lactobacillus plantarum YW-11 are crucial for significant probiotic activity, play a role as these enzymes build and break down complex carbohydrates and glycoconjugates for a large body of biological roles.

Regards

Yang Zhennai (Professor)

School of Food and Health

Beijing Technology and Business University, Beijing China

Reviewer 2 Report

The manuscript Genome investigation and Functional Annotation of Lacti- plantibacillus plantarum YW11 Revealing Streptin and Ruminococcin-A as Potent Nutritive Bacteriocins against Gut Symbiotic Pathogens describes that an advanced analysis to study the interaction of probiotics with the gastrointestinal environment, irritable bowel disease, and immune responses along with the analysis of the secondary metabolites’characteristics of Lp YW11.

It can provide value information in the food safety approaches and in the food starter cultures.

This study is within the scope of Molecules. The experimental design was reasonable and diversified., and the paper contains interesting results. The presented manuscript could be considered for publication after minor revision as following mentioned: 

(1) Line 115 Previously isolated Lactiplantibacillus plantarum YW11 (Lp YW11), the strain number need to be supplemented.

(2) The clarity of Figure 1 and Figure 3 is not good and needs to be adjusted.

(3) It is suggested to modify several illustrations in the text. For example, Figure 2 shows …… ,that this statement is not appropriate.

(4) There are a few text and formatting errors in the text, it is recommended to modify. 

Author Response

Dear Editor and Reviewers,

We thank you for your critical review of our manuscript and for giving us the chance to submit the revision. The comments were constructive, and we tried to address all of them. We have attached the revised manuscript and a specific response to all three reviewers. We also highlighted the changes in the manuscript. We hope that the manuscript will now be considered suitable for publication.

Reviewer 2

The manuscript “Genome investigation and Functional Annotation of Lactiplantibacillus plantarum YW11 Revealing Streptin and Ruminococcin-A as Potent Nutritive Bacteriocins against Gut Symbiotic Pathogens” describes that an advanced analysis to study the interaction of probiotics with the gastrointestinal environment, irritable bowel disease, and immune responses along with the analysis of the secondary metabolites’ characteristics of Lp YW11. It can provide value information in the food safety approaches and in the food starter cultures.

Response: Thank you for your appreciation. It means a lot to us.

This study is within the scope of Molecules. The experimental design was reasonable and diversified., and the paper contains interesting results. The presented manuscript could be considered for publication after minor revision as following mentioned: 

(1) Line 115 “ Previously isolated Lactiplantibacillus plantarum YW11 (Lp YW11)”, the strain number need to be supplemented.

Response: Thank you for your comment. The strain number has been supplemented in the revised manuscript. Please see the revised manuscript [Lines 115-116].

(2) The clarity of Figure 1 and Figure 3 is not good and needs to be adjusted.

Response: Thank you for your comment. Figure 1 and Figure 3 have been adjusted.

(3) It is suggested to modify several illustrations in the text. For example, “Figure 2 shows ……” ,that this statement is not appropriate.

Response: Thank you so much for your comment. It has been changed accordingly.

(4) There are a few text and formatting errors in the text, it is recommended to modify. 

Response: Thank you for your comment. The text and formatting errors are modified in the revised manuscript.

 Regards

Yang Zhennai (Professor)

School of Food and Health

Beijing Technology and Business University, Beijing China

Round 2

Reviewer 1 Report

Review comments

Revision has been done as mentioned in the first review comments. English correction has also been done in most of the part of the manuscript as mentioned previously in the first review comment. However, the same issues persist in the revised manuscript, particularly with the used bioinformatics tools. I don't find the author's argument to be convincing. This manuscript, in my opinion, does not meet to the standards of this Journal Molecules for publication.

Though the authors have tried making some changes in the manuscript the revised version is not convincing.

1.      Previously, the authors were asked to make changes in the figure by making their own graph rather than snipping. However, no changes were made. The authors should make their own graph in case of figure 1, 4, 5, and 6.

2.      The figure 2 is too vague and difficult to comprehend the readers, so either it can be moved to the supplementary file or removed from the manuscript. The authors are trying to show the carbohydrate and secondary metabolites metabolic pathway, It would be better if they create a table and list the pathways and genes involved as a supplementary table rather than the figure representation.

3.      The explanation to the previously raised question is not enough, “The glycoside hydrolase and glycosyltransferase families of carbohydrate-active enzymes, which are largely contributed to by the genome of Lp YW11, are crucial for significant probiotic activity and the control of the immune response to a variety of pathogens.” Please explain how the GH and GT families present in YW11 strain are crucial for probiotic activity. Here the authors should explain in detail and provide references about how GT and GH families found in the strain have shown such role as probiotic and control the immune response and mention in the discussion section of the manuscript.

In addition, there are some places, which have not even been corrected even after the first review comment.

1.      In page no 2, line no 83 and 84, all the genus names should be in italics in scientific writing. So please correct the mistake e.g.; Salmonella, Shigella, Campylobacter etc.

2.      Figure 8, Collinearity relationship of YW-11, please make the text and value more bigger as it is not clear and difficult to read.

3.      In line no 116 of materials and method section please write (-9 oC) instead of (-9 oC).

Author Response

Dear Editors and Reviewers

We sincerely thank you for your valuable comments, we have revised the manuscript according to your valuable comments. All the changes have been highlighted in the revised manuscript. We hope that now the article will be considered for publication.

Reviewer 1 (Round 2)

Revision has been done as mentioned in the first review comments. English correction has also been done in most of the part of the manuscript as mentioned previously in the first review comment. However, the same issues persist in the revised manuscript, particularly with the used bioinformatics tools. I don't find the author's argument to be convincing. This manuscript, in my opinion, does not meet to the standards of this Journal Molecules for publication.

Though the authors have tried making some changes in the manuscript the revised version is not convincing.

  1. Previously, the authors were asked to make changes in the figure by making their own graph rather than snipping. However, no changes were made. The authors should make their own graph in case of figure 1, 4, 5, and 6.

AR: Thank you so much for your comment. The requested changes in figures have been made please see revised manuscript.

  1. The figure 2 is too vague and difficult to comprehend the readers, so either it can be moved to the supplementary file or removed from the manuscript. The authors are trying to show the carbohydrate and secondary metabolites metabolic pathway, It would be better if they create a table and list the pathways and genes involved as a supplementary table rather than the figure representation.

AR: Thank you for your comment. The figure 2 has been moved to supplementary data.

  1. The explanation to the previously raised question is not enough, “The glycoside hydrolase and glycosyltransferase families of carbohydrate-active enzymes, which are largely contributed to by the genome of Lp YW11, are crucial for significant probiotic activity and the control of the immune response to a variety of pathogens.” Please explain how the GH and GT families present in YW11 strain are crucial for probiotic activity. Here the authors should explain in detail and provide references about how GT and GH families found in the strain have shown such role as probiotic and control the immune response and mention in the discussion section of the manuscript.

 AR: Thank you for your comment. Probiotic microorganisms exert their effects in a number of ways, such as by modifying immune response, generating organic acids and antimicrobial compounds, interacting with the host's microbiota, enhancing gut barrier integrity, and producing enzymes. Probiotics come in a variety of flavours. They are mostly oligosaccharide carbohydrates and belong to a subset of carbohydrate groups (OSCs). The formation of a sugar hemiacetal or hemiketal and the corresponding free aglycon is caused by the hydrolysis of the glycosidic linkage of glycosides, which is catalysed by glycoside hydrolases and glycosyltransferases.

In addition, there are some places, which have not even been corrected even after the first review comment.

  1. In page no 2, line no 83 and 84, all the genus names should be in italics in scientific writing. So please correct the mistake e.g.; Salmonella, Shigella, Campylobacteretc.

AR: Thank you for pointing out this mistake. Throughout the manuscript all the species names have been written in Italic in the revised manuscript.

  1. Figure 8, Collinearity relationship of YW-11, please make the text and value more bigger as it is not clear and difficult to read.

AR: Thank you for pointing out this mistake. Figure 8 has been revised.

  1. In line no 116 of materials and method section please write (-9 oC) instead of (-9 oC).

AR: Thank you for pointing out this mistake. It has been corrected. Please see revised manuscript.

Regards

Professor Dr. Yang Zhennai

School of Food Science & Health

Beijing Technology & Business University

Haidian District Beijing China
